# Towards Smaller and Stronger: An Edge-Aware Lightweight Segmentation Approach for Unmanned Surface Vehicles in Water Scenarios

**DOI:** 10.3390/s23104789

**Published:** 2023-05-16

**Authors:** Wei Han, Binyu Zhao, Jun Luo

**Affiliations:** 1School of Mechatronic Engineering and Automation, Shanghai University, Shanghai 200444, China; 2Systems Engineering Research Institute, China State Shipbuilding Corporation, Beijing 100094, China; 3School of Computer Science and Techonogy, Harbin Institute of Technology, Harbin 150001, China

**Keywords:** water scenario segmentation, edge awareness, lightweight networks, unmanned surface vehicles

## Abstract

The accurate detection and segmentation of accessible surface regions in water scenarios is one of the indispensable capabilities of surface unmanned vehicle systems. ‘Most existing methods focus on accuracy and ignore the lightweight and real-time demands. Therefore, they are not suitable for embedded devices, which have been wildly applied in practical applications.‘ An edge-aware lightweight water scenario segmentation method (ELNet), which establishes a lighter yet better network with lower computation, is proposed. ELNet utilizes two-stream learning and edge-prior information. Except for the context stream, a spatial stream is expanded to learn spatial details in low-level layers with no extra computation cost in the inference stage. Meanwhile, edge-prior information is introduced to the two streams, which expands the perspectives of pixel-level visual modeling. The experimental results are 45.21% in FPS, 98.5% in detection robustness, 75.1% in F-score on MODS benchmark, 97.82% in precision, and 93.96% in F-score on USV Inland dataset. It demonstrates that ELNet uses fewer parameters to achieve comparable accuracy and better real-time performance.

## 1. Introduction

Safe sailing is the premise for unmanned surface vehicles (USVs) to carry out diverse tasks [1]. Scenario perception and understanding is a fundamental capability of the unmanned system. Detecting, localizing and recognizing objects or obstacles is one of the functions for ensuring safe inland and maritime autonomous sailing, which is as vital as unmanned aerial vehicles (UAVs) and unmanned ground vehicles (UGVs). Considering the effectiveness and the cost, the camera is the most information-dense, affordable sensor for water scenario understanding [2]. Additionally, detecting accessible regions in visual images is always a central issue for safe navigation in water scenes.

Water scenario segmentation aims to distinguish the water and obstacles in inland or maritime scenarios, which provide the situation awareness and notations for safe driving of USVs. Generally, it has more strict requirements on accuracy and efficiency on segmentation algorithms for unmanned systems.

In the past decades, some researchers attempted to utilize horizon line extraction methods to solve this problem [3,4,5,6]. However, they are not suitable for varied water scenarios, such as the situation of interception or disaster relief in maritime environments, or the situation of water quality monitoring or waste management inland. Therefore, a reliable solution for situation awareness and scenario segmentation in such immensely supplemented and expanded water scenarios is important for autonomous navigation of USVs.

In recent years, convolutional neural networks (CNNs) have been proven an effective approach applying to various fields, which provide rich deep features with excellent perception results on different platforms. Various powerful CNN-based water scenario segmentation approaches have been proposed [7,8,9,10,11,12,13,14]. Compared with traditional approaches, these methods achieve better performance. However, most of those methods do not focus on the number of parameters, which affects the performance of the model. Water scenario segmentation aims to give real-time supports for downstream tasks, such as obstacle avoidance in harbors or floating waste detection in inland rivers. To cope with the complex environment or strict requirements, especially for situations such as USVs or devices embedded with multiple task-specific software within limited computation resources, the processing speed of existing water scenario segmentation methods still needs improving. Thus, a lighter segmentation network is required to further compress the computational cost.

On the other hand, water scenario segmentation detects the boundary between water, obstacles, sky regions in some extent. Edge information is a powerful prior information in both visual data and manual annotations. As one of the traditional segmentation approaches, edge features are still one of the popular strategies in current deep learning methods. For example, Lee et al. [15] utilize an edge map as one of the inputs to capture rough semantics; and Lu et al. [16] and Fan et al. [17] utilize an auxiliary boundary-aware stream extracted from the ground truth to make salient features and further estimate the silhouette and segmentation of objects. Inspired by those approaches, we are conscious of its importance in raw image and ground truth data. It is rarely adopted in existing water scenario segmentation methods. Therefore, we establish a spatial-stream learning integrating with the edge information.

In this paper, we propose an edge-aware lightweight water scenario segmentation method (dubbed as ELNet) for USV systems, which reduces the structure complexity and enhances the perception capability for water scenarios. Specifically, we build a two-stream learning strategy consisting of a context stream and a well-designed spatial stream. First, edge-prior information is concatenated to the context stream, the purpose of which is to reduce the parameters in the context stream and fully utilize the edge similarity between the raw image and ground truth. Second, we design an edge-dominant spatial stream, which only works on the training stage, to assist the feature learning with no parameter introduction. Edge-prior information is encoded and coupled with the ground truth to guide the detail feature learning. These designs normalize and enrich the model features, including not only the edge-related but also inter-class granularity semantics in the raw data. The main contributions are summarized as follows:We propose a light segmentation method for water scenarios, which utilizes a two-stream learning strategy. Except for the traditional context stream, a spatial stream is expanded to learn spatial details in low-level layers with no extra computation cost in the inference time.We introduce edge-prior information to different layers in both streams, which leads to object-level semantic learning and memorizing, and expands the perspectives of pixel-level visual modeling.Evaluation on MODS benchmark and USV Inland dataset demonstrate that our approach achieves compelling performance. Notably, we obtain a significant improvement with a much lower number of parameters than the best frame-grained method.

This paper is organized as follows. In Section 2, the related works of existing water scenario segmentation methods, edge detection and lightweight networks are introduced. Then, in Section 3, we present how the proposed network ELNet works and its detail designs. In Section 4, experimental settings and results validate the performance of the proposed approach. Finally, the work is concluded briefly in Section 5.

## 2. Related Works

### 2.1. Water Scenario Segmentation

Semantic segmentation is a classic problem in the field of computer vision. In the semantic segmentation problem, each pixel in an image is assigned with a category ID according to the object of interest to which it belongs [18,19,20]. Recently, segmentation in water scenarios also attracts much attention due to the development of unmanned surface vehicle systems. Water scenario segmentation aims to semantically classify each pixel in images to obstacles/environment region, water region and sky region. The segmentation provides an accurate separation on non-water regions, so that fully autonomous navigation of USVs is realized in complex water surfaces.

In recent years, vision-based segmentation methods have achieved promising improvements modeling with CNN [8,9,10,11,12,13,14]. As for inland waterways, Bovcon et al. extend a single-view model to a stereo system, and finally propose a stereo-based obstacle detection method [8]. Zhou et al. propose an inland collision-free waterway segmentation model by obtaining pixel-wise classification [10]. Vandaele et al. improve inland semantic results on two segmentation datasets by applying transfer learning to segmentation [11].

As for maritime scenarios, Chen et al. propose an attention-based semantic segmentation network through designing an attention refine module (ARM) to improve detection accuracy at sea–sky line areas [9]. Kim et al. propose a vision sensor-based Skip-ENet model to recognize marine obstacles effectively within a limited amount of computation cost [12]. Bovcon et al. propose a water-obstacle separation and refinement network (WaSR) to improve the estimation of the water edge, detection of small obstacles and high false-positive rates on water reflections and wakes [13].

### 2.2. Edge Detection

Edge detection can localize and extract significant variations (the boundary between different objects) in an image, which benefits various vision-based tasks. Bertasius et al. utilize a multi-scale deep network to exploit object-related features as high-level cues for contour detection [21]. Cooperated with feature fusion, Yu et al. propose a novel skip layer design and a multi-label loss function for semantic edge detection tasks [22]. Shen et al. adapt the training strategy with edge information, in which contour data are portioned into sub-classes and each subclass is fitted by different model parameters [23].

In this paper, different applications of edge information are explored to benefit the model learning, feature analysis, and further successfully improve the performance on pixel-level predictions.

### 2.3. Lightweight Networks

CNN has applications on diverse research fields, and thus the performance on many tasks is tremendously improved. However, one of the following critical questions is efficiency. Conventional CNN reasoning is quite difficult to apply on resource constrained scenarios, such as mobile terminal and Internet of Things. The reason is that CNN requires a large amount of computation. Only through complex tailoring that CNN models could be reluctantly deployed to the mobile end. Fortunately, starting from SqueezeNet [24] and MobileNetV1 [25], researchers gradually pay attention to the efficiency problems in resource constrained scenarios. After several years of development, the relatively mature lightweight networks include MnasNet [26], mobilenet series [25,27,28], Interception series [29,30,31], FBNet series [32,33,34], and GhostNet [35].

MobileNetV2 refers to the residual structure of ResNet and introduces the inverted residual module for further improvements. On the one hand, the residual structure is conducive to the network’s learning. It reduces the calculation amount of the original point-wise convolution. Although MobileNetV3 learns from the advantages of MnasNet and MobiNet series, the improvement mainly contributes from manual efforts, which are not flexible enough to serve as the backbone. In this paper, MobileNetV2 is selected as the encoder backbone of the proposed network.

## 3. Method

The overview of the proposed edge-aware lightweight network (ELNet) is shown in Figure 1. First, edge detection is conducted to acquire the edge information, which also serves as the input data of the network. Then, the raw sensor data and corresponding edge data are input into the context stream and spatial stream. The context stream utilizes the classic encoder–decoder structure to inference the segmentation result, and the spatial stream implements the feature guidance mainly in the low-level layer. In this section, we introduce the details of the proposed network, and describe the whole architecture in detail.

### 3.1. Network Architecture

**Encoder.** ELNet follows the traditional encoder–decoder structure to obtain the segmentation result. The encoder in the context stream is extremely essential for feature extraction and latent feature analysis. We choose MobileNetV2 [27] as the backbone, and most of the original design in the network will be retained as possible. Except for the last module for classification, MobileNetV2 has a total of 17 calculation modules. It contains multiple bottleneck blocks, which vary in five scales to extract features from the original image progressively. 

To achieve a deep perception of the edge, edge-prior information is concatenated to each encoding stage. A small fraction of the feature channels are chosen to store the boundary semantics, which also slightly reduce the number of parameters. Considering to match the size of the learning features, the average pooling is selected to align the size of the edge feature for fusion. We generate binary edge information from the raw sensor data by Laplacian Operator $k_{laplace}$, and the parameter is set as
(1)$$k_{laplace}=\left(\begin{array}{ccc} 0 & 1 & 0\\ 1 & -4 & 1\\ 0 & 1 & 0 \end{array}\right)$$

We select this basic filter mask as the parameters of the utilized Laplacian Operator, because it has the most generalization applied to almost all cases for edge detection.

**Decoder.** To minimize the total number of parameters, we utilize the representative feature maps in three stages: the first stage, the third stage and the last stage. It is believed that the features in the first stage preserve the fundamental pixel-level information such as shapes of objects, the features in the last stage preserve the most abstract semantic information, and the features in the third stage are guided by the edge detail from the inputs and ground truth. Therefore, the features in the second and fourth stage are not as vital as the features in the first, third and last stages. For the feature maps at the first and third stage, the features after the last convolution layer is abandoned, instead by the features when the number of channels keeps at the maximum. This selection is conducive to fully preserving enriching information obtained by the encoder, and contributes to favorable information for segmentation. We also design an ablation study to validate the rationality of this design, and the experimental results are given at Section 4.7. The decoding and fusion strategy in the decoder can be formulated as:(2)$$\begin{array}{cc} \; & {\mathrm{fea}}_{i}=TransposeConv\left({\mathrm{fea}}_{i}\right)\\ \; & {\mathrm{fea}}_{i}=Normalization\left({\mathrm{fea}}_{i}\right)\\ \; & {\mathrm{fea}}_{i}=Dropout\left({\mathrm{fea}}_{i}\right)\\ \; & {\mathrm{fea}}_{i}=Activation\left({\mathrm{fea}}_{i}\right)\\ \; & {\mathrm{fea}}_{i-2}=Concatenate\left({\mathrm{fea}}_{i},{\mathrm{fea}}_{i-2}\right)\\ \; & \left(i=3,\;5\right) \end{array}$$
where ${\mathrm{fea}}_{i}$ denotes the feature to-be-processed in the *i*-th stage. Additionally, batch normalization [36] is used for normalization and ReLU function [37] for activation in this paper. The dropout rate is set as 0.5 [38].**Auxiliary detail guidance.** As shown in Figure 1, we use the low-level features (the 3rd stage) to produce detail guidance via the edge information and segmentation ground truth. Specifically, the guidance comes from two perspectives: consistent with the features of edge-prior information calculated from the Laplacian operator and with the ground truth of segmentation after decoding. Based on the above description, a Convolution Head is raised to regularize the third-stage feature maps. This module is carried out by
(3)$$\begin{array}{cc} ConvHead(I)= & Conv_{2}(Activation(\\ \; & Normalization(Conv_{1}\left(I\right)))) \end{array}$$
where I denotes the input features, and the “Normalization” and “Activation” are batch normalization and ReLU function, respectively. Additionally, the kernels of the two convolution layer are 3×3 and 1×1, respectively.

The edge features ${\mathrm{fea}}_{edge}$ and distinguished features ${\mathrm{fea}}_{jdg}$ are defined as:(4)$$\begin{array}{cc} \; & {\mathrm{fea}}_{edge}=ConvHead1\left(\mathrm{LD}\right)\\ \; & {\mathrm{fea}}_{jdg}=ConvHead2\left({\mathrm{fea}}_{3}\right) \end{array}$$
where “LD” is the Laplacian Detection input image, and ${\mathrm{fea}}_{3}$ denotes the entire third-stage features after processing.

Note that this stream is discarded in the inference phase. Therefore, the spatial stream can boost the accuracy of prediction and introduce no additional parameter at the same time.

**Network details.** Table 1 shows the detailed structure of the proposed network ELNet.

### 3.2. Detail Guidance in Spatial Path

Inspired by [17,39], compared with a single stream network, which provides the context information on the original backbone’s low-level layers (the 3rd stage). An additional spatial stream can encode more spatial detail for complementary, e.g., boundary, and corners. Based on this observation, we utilize the auxiliary stream to guide the low-level features to learn the spatial information independently.

**Guide with edge prior information.** The detail feature prediction is modeled as a small knowledge distillation task and a binary segmentation task. We first generate the edge features encoded from the input edge image by a Laplacian operator and guide the partial learning of the third-scale coding features, which learns the same information from input image pairs. It can be illustrated as
(5)$$L_{fea}=L_{1}\left({\mathrm{fea}}_{edge},{\mathrm{fea}}_{3}\left[N,:,:\right]\right)$$
where $L_{1}$ is the L1 loss. Specifically, the last N channel features are guided to be consistent with the knowledge from the input edge image.**Guide with ground truth.** Then, another Conv Head is utilized to generate segmentation prediction with the whole third-stage feature map and the detail ground-truth, which guides the feature map of the low-level layer to learn more spatial details. As shown in Figure 1, this guidance can be formulated as
(6)$$L_{3rd-seg}=L_{focal}\left(p_{s},g_{s}\right)$$
where $p_{s}\in R^{H\times W}$ denotes the pixel-level spatial features ${\mathrm{fea}}_{jdg}$ and $g_{s}\in R^{H\times W}$ denotes the corresponding downscale spatial ground truth, where the downscale factor is 8. $L_{focal}$ is the focal loss with cross entropy modified from [40].

### 3.3. Total Loss Function

The total loss function of the proposed network ELNet is composed of three categories:(7)$$L=\lambda_{fea}L_{fea}+\lambda_{3rd-seg}L_{3rd-seg}+\lambda_{seg}L_{seg}$$
where $\lambda_{i}$ is corresponding weight to balance the three items. The first two items, $L_{fea}$ and $L_{3rd-seg}$, are stated in Section 3.2. The last item $L_{seg}$ is also the focal cross-entropy loss at the original image scale between the final prediction and ground truth. Though there is not the class-imbalance situation that the number of positive pixels (water) is not much less than the negative pixels (non-water), the reason to use focal loss is the attempt on a stronger penalty on a false prediction. The focal cross entropy loss is written as
(8)$$L_{focal}=-\alpha {\left(1-p\right)}^{\gamma }log(p)$$
where γ is the focal rate, α is a balancing parameter, and p∈[0, 1] describes the probability of the predicted pixels, which is defined as
(9)$$p=exp(-L_{CE})$$

In this paper, the weight group is set as $\lambda_{fea}=1,\lambda_{3rd-seg}=1$ and $\lambda_{seg}=1$ in the subsequent experiments. For focal cross-entropy loss, the setup γ=2,α=0.25 follows [40].

## 4. Experiments

### 4.1. Dataset and Benchmark

Following the evaluation method of Bovcon et al. [41], we use the MaSTr1325 public dataset for training, and validate the proposed method on the MODS benchmark and USVInland dataset.

**MaSTr1325** [42]: Marine Semantic Segmentation Training Dataset (MaSTr1325) is specially used to develop obstacle detection methods for small coastal USVs. The dataset contains 1325 reality-captured images, which include obstacles, water surface, sky and unknown targets, covering a series of real conditions encountered in coastal surveillance missions. It captures a variety of weather conditions, which range from foggy, partly cloudy with sunrise, overcast to sunny, and visually diverse obstacles, which are shown in Figure 2. The image size of MaSTr1325 is 512×384.**MODS benchmark.** [41]: The goal of MODS is to benchmark segmentation-based and detection-based obstacle detection methods for the maritime domain, specifically for use in unmanned surface vehicles (USVs). For segmentation-based detection, the segmentation method classifies each pixel in a given sensor image into one of three classes: sky, water or obstacle. Additionally, the MaSTr1325 dataset was created specifically for training. 

MODS benchmark totally consists of 94 maritime sequences, which consist of approximately 8000 annotated frames with over 60k annotated objects. It consists obstacles annotation in two types: dynamic obstacles, which are objects floating in the water such as boats and buoys; and static obstacles, which are all remaining obstacle regions such as shorelines and piers.

**USVInland** [43]: Different from the condition on the sea, the inland river environment, which is relatively narrow and complex, often brings additional challenges to the positioning and perception of the USV. Compared to the emerged public datasets in the field of road automatic driving, such as KITTI [44], Oxford RobotCar [45] and nuScenes [46], the USVInland dataset undoubtedly fills the gap and opens a new situation for inland river unmanned ships. A total of 27 pieces of original data are collected. There are relatively low resolution (640×320) and high resolution (1280×640) images in the water segmentation sub-dataset. The fully original data will be directly used for validation.

### 4.2. Evaluation Metrics

Different from general segmentation metrics, MODS score models use USV-oriented metrics. They focus on the obstacle detection capabilities of methods. For dynamic obstacles, MODS annotates them with bounding boxes. For static obstacles, MODS labels the boundary between static obstacles and water annotated as polylines. In this paper, we utilize water-edge accuracy ($\mu_{A}$) and detection robustness ($\mu_{R}$) to evaluate the capability of the baseline and the proposed method to detect water edges, and precision (Pr), recall (Re), and F-score to evaluate the accuracy of segmentation.

For USVInland dataset, water-edge polylines will be annotated manually. Precision (Pr), recall (Re) and F-score are also chosen to evaluate the segmentation performance. The formulas of the metrics are as follows:(10)$$\begin{array}{cc} \; & \mu_{A}=\frac{\sum_{i=0}^{N_{frame}}WE-RMSE_{i}}{N_{frame}}\\ \; & \mu_{R}=\frac{TP_{land}}{TP_{land}+FP_{land}}\\ \; & Pr=\frac{TP}{TP+FP}\\ \; & Re=\frac{TP}{TP+FN}\\ \; & F-score=2\times \frac{Pr\times Re}{Pr+Re} \end{array}$$
where $WE-RMSE_{i}$ denotes the water edge RMSE of *i*-th frame.

### 4.3. Implementation Details

Though MaSTr1325 dataset holds various weather conditions, data augmentation is also recommended by officials to make up the quantity of the dataset, prevent overfitting and simulate diverse cruising conditions. Therefore, a series of augmentation methods containing random horizontal flipping, random rotating by up to 15 degrees, random scaling from 60 to 90 percent, and color changes are adopted with 50%, 20%, 20%, and 20% possibilities when training with MaSTr1325, which is more strict than implementations in other papers and poses challenges to existing methods. Additionally, since the resolution is not consistent in USVInlnad dataset, high resolution images will be resized to low resolution.

All approaches are trained with the PyTorch framework [47] and the Adam optimizer [48] using GeForce GTX 1080 Ti. The learning rates are all initially set as $1\times 10^{-4}$ and are halved for every 25 epochs. The values of the input images are normalized within the range [0, 1]. Additionally, the implementation of ELNet is imported from the pytorch library, which also follows the preprocess in pytorch that a special mixture of mean=[0.485, 0.456, 0.406] and std=[0.229, 0.224, 0.225] normalization are operated before training.

### 4.4. Comparison with Related Segmentation Methods

In the experiments, state-of-the-art algorithms are compared with the proposed network: WaSR [13], WODIS [9], CollisionFree [10], Skip-ENet [12], ShorelineNet [14] and a general-purpose segmentation network BiSeNet [39]. The model of WaSR, Skip-ENet and BiSeNet are obtained from official code, and the backbone of WaSR we choose is ResNet101 with no IMU. The model of WODIS and CollisionFree are reproduced by ourselves based on their published papers.

The parameter levels and inference speed of these methods are collected in Table 2. The total inference time has already contained the time consumed on acquiring the edge prior. In fact, it only consumes a little time, which can be neglected. It is referred that the proposed network ELNet has an extremely evident advantage on the total number of parameters, ∼4.86 M. Except for the lightest model Skip-ENet, ELNet is 26% smaller than ShorelineNet, and far lighter than the state-of-the art methods WaSR and good general segmentation methods BiSeNet. This achieves the goal of establishing a lightweight network.

As for inference speed, ELNet ranks higher in the compared group both on GPU and CPU, which is much lower than the state-of-the-art methods WaSR (10.63 s/0.98 s on GPU/CPU) for water scenario and BiSeNet (36.12 s/4.87 s on GPU/CPU) in general. However, the qualititative results of Skip-ENet and ShorelineNet are unstable according to Table 3, which is unfavorable for scenario perception. The improvement on inference speed mainly comes from three aspects:First, a two-stream learning strategy (context stream learning with spatial stream learning) is applied. Additionally, the spatial stream works only in the training stage, which reduces the computation cost and thus affects the speed in the inference time.Second, the backbone of the proposed network refers to the designs of lightweight networks such as MobileNet series, Interception serires, etc., which have been approved to have a faster speed than traditional CNN networks.Third, we select an asymmetric decoder rather than a symmetric one with encoder after experiments, which also contributes to the speed in the inference time. The experimental detail is discussed in Section 4.7.

### 4.5. Performance on the MODS Benchmark

The training results of ELNet and qualitative evaluation results on MODS benchmark are shown in Figure 3. The convergent trend in the training curve shows that ELNet has successfully modeled the semantics of the scenario in the MaSTr1325 dataset. The training loss curves of ELNet on the MaSTr1325 dataset confirms that the model has converged rapidly in the first 5000 iterations. From our point of view, the reasons that a couple of peaks emerged in the loss curve after the second 5000 iterations are: (a) the scene of the sampled data is challenging to the model in the training stage so that the model performs a little worse than models of neighboring iterations, such as the sky–water line is long and hard to distinguish in the sampled data; and (b) strict data augmentation including random horizontal flipping, random rotating, random scaling, and color changes is adopted, which also intensify the difficulty. In fact, almost all the loss keeps the difference within 0.05 after the second 5000 iterations, which is far smaller than the difference (>0.8) in the first 5000 iterations. Combining with the training accuracy curve, we regard it as a normal phenomenon.

Among all experimental results in Figure 4, the edge detection result of WODIS and the obstacle detection result of ShorelineNet are the worst, while the results of WaSR, BiSeNet and ELNet reveal more accurate sea edge detection and positive obstacle detection performance. Additionally, for segmentation results, it is observed in the first row that WODIS and ShorelineNet are the left over most-negative pixels, while BiSeNet and ELNet have less classification errors for each pixel. Especially for identifying obstacles, the proposed network ELNet achieves the better segmentation with contours, which is evidently shown in the yellow circle.

The quantitative evaluation results are summarized in Table 3. The table shows that WaSR and the proposed method ELNet achieve a much higher water edge accuracy (11 px for both WaSR and ELNet). Except for the water-edge detection, CollisionFree achieves the best precision with 75% and F-score with 80.7%, but fails on recall with 86.7%. Meanwhile, WaSR has the best accuracy on recall with 98.3%, but the precision and F-score are both lower than ELNet. Among the whole metrics, ELNet reveals the best trade-off within the best water-edge detection and other segmentation results. Meanwhile, ELNet has a much lower number of parameters and better performance in qualitative evaluation as aforementioned, which reach the original pursuit for a smaller and stronger model.

### 4.6. Performance on USVInland Dataset

The utilized model on USVInland is also training on the MaSTr1325 dataset, the purpose of which is to evaluate the capability of transfer learning. The experimental results are illustrated in Figure 5 and Table 4. Since the ground truth does not distinguish the sky and obstacles, we have to compare the quality of obstacle–water segmentation and the water–sky-line only. From Figure 5, we observe that WaSR and ELNet achieve more accurate edge detection through labeling the obstacle–water edge manually. CollisionFree fails to perceive the boundary of water the in river scenarios. For the segmentation result, ELNet has less false positive pixels on detecting water regions.

From Table 4, the results of the proposed network ELNet reveals a comparable performance with the state-of-the-art method WaSR. This observation confirms that ELNet achieves similar performance compared to other excellent water segmentation approaches with a lighter network, which proves the strong robustness of the proposed network. Therefore, it is believed that ELNet has more competence performing well when transferring the model to other dataset.

### 4.7. Ablation Study on the Number of Upsampling Blocks

Five types of links between the encoder and decoder are trained on the MaSTR1325 dataset and validated on the MODS benchmark following the implementation details in Section 4.3, which are summarized as:Type 1: features in stage 1, 5, which preserve the features of the highest- and lowest-level.Type 2: features in stage 3, 5, which preserve the features of the finest-guidance and lowest-level.Type 3: features in stage 1, 3, 5, which preserve the features of the highest-level, lowest-level and the finest-guidance.Type 4: features in stage 1, 2, 4, 5, which preserve the features apart from that of the finest-guidance.Type 5: features in stage 1, 2, 3, 4, 5, which classically preserve the features of all levels.

The experimental results are shown in Table 5. The comparison between Type 2 and Type 3 demonstrates the importance of low-level layer features on fundamental cognition. additionally, the results from Type 1 and Type 3 validate the improvements of the low-level features in the third stage. Type 4 has one large upsample kernel (the kernel size is 4) instead of two small kernels, while it achieves better performance than Type 5.

It is observed that the model of Type 4 has a comparable performance with that of Type 3, which is the second best structure type. Although it is evident that Type 3 has nearly identical performance compared to Type 4, the number of total parameters is 20% lower than Type 4. This is the basic judgment the proposed network ultimately adopts.

### 4.8. Ablation Study on Edge-Aware Modules

We define the special designed components in ELNet as *Fusion* and *Auxiliary*, where *Fusion* means the concatenation of the edge prior calculated through a Laplacian operator and the main branch feature maps. *Auxiliary* means the entire spatial stream, which includes two convolutional heads and their corresponding objective functions. The experiments about the effectiveness of the two components also follow the paradigm that the models are trained on the MaSTR1325 dataset and validated on the USVInland dataset with the implementation details in Section 4.3. Table 6 illustrates the final results.

It can be observed from the first row and the second row that the *Fusion* strategy helps better recognize objects in visual data. The same case also happens on the *Auxiliary* strategy when comparing the metrics in the first row with the third row, and have a larger margin than the *Fusion* strategy. This demonstrates that the spatial stream contributes more to the improvements. When the two strategies are implemented simultaneously, the final result naturally achieves the best. Nevertheless, it is worthy of noting that ELNet barely has 88.69% on recall and 93.96% on precision, and lower false segmentation is an advantage when considering a safer navigation on the water surface in maritime environments. This is a key point that needs to be improved in the future.

## 5. Conclusions

In this paper, an edge-aware lightweight algorithm, ELNet, is proposed to promote practical development for unmanned surface systems. By leveraging two-stream learning, the proposed method achieves better perception on object-level details with limited computation cost. By taking a strongly edge-guided optimizing direction in the two streams, ELNet achieves a visible margin on the comprehensive accuracy of segmentation. In addition, the generalization capability of ELNet is validated by training and testing on different maritime datasets. In particular, ELNet achieves a more than 50% parameter reduction, and holistically stable performance in detection accuracy (i.e., precision, recall and F-score pair) in comparison with the state-of-the-art methods *WaSR* [13] and *BiSeNet* [39], which suggests that it can provide safer passable regions for USVs. The excellent robustness and stability also demonstrate the potential for application in more water scenarios.

## Figures and Tables

**Figure 1 sensors-23-04789-f001:**
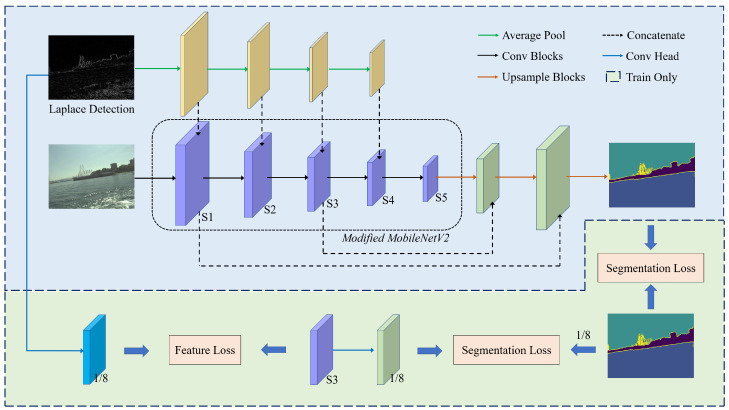
Overview of the ELNet segmentation algorithm. The ELNet consists of two streams, context stream and spatial stream. The backbone of context stream is selected as MoblieNetV2 [27], and “S1–S5” represent the different stages of intermediate context features in different resolutions. The resolution of “S1” is the largest, which is half the size of the input image, and the resolution of “S5” is the smallest, which is 1/32 the size of the input image.

**Figure 2 sensors-23-04789-f002:**
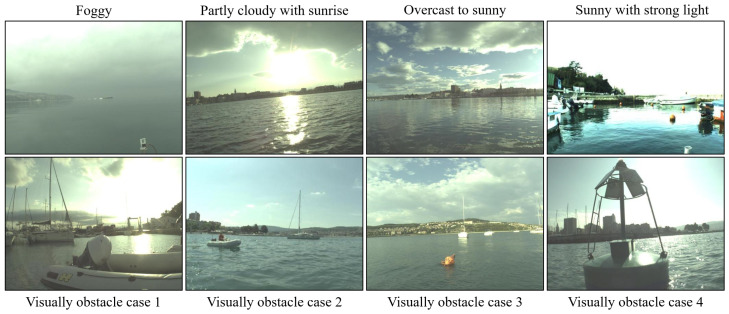
The MaSTr1325 contains a series of weather conditions and diverse object/obstacles, which provides a broad range of appearances and types for scenario segmentation.

**Figure 3 sensors-23-04789-f003:**
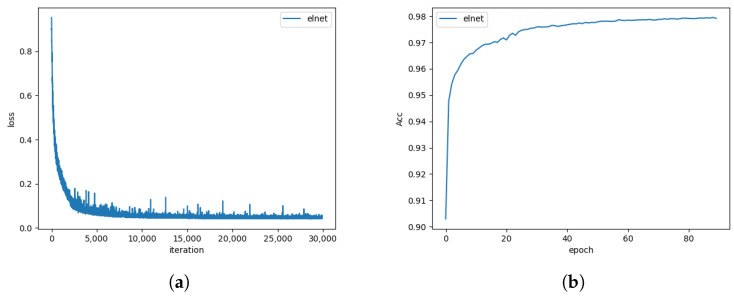
Training loss (**a**) and accuracy (**b**) results of ELNet on MaSTr1325 dataset.

**Figure 4 sensors-23-04789-f004:**
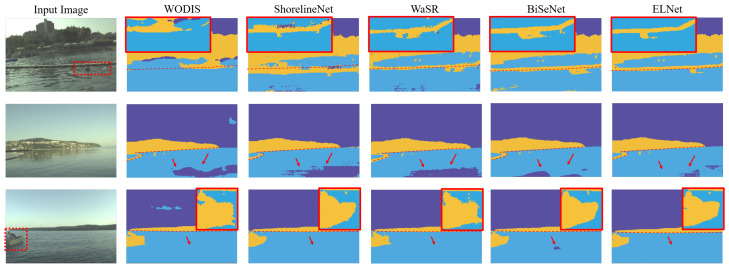
Qualitative evaluations on the MODS benchmark. The sky, obstacles and water are colored in deep-blue, yellow and cyan color, respectively. The red dashed line indicates the ground truth sea edge, and the main differences are outlined with arrows or zoomed in.

**Figure 5 sensors-23-04789-f005:**
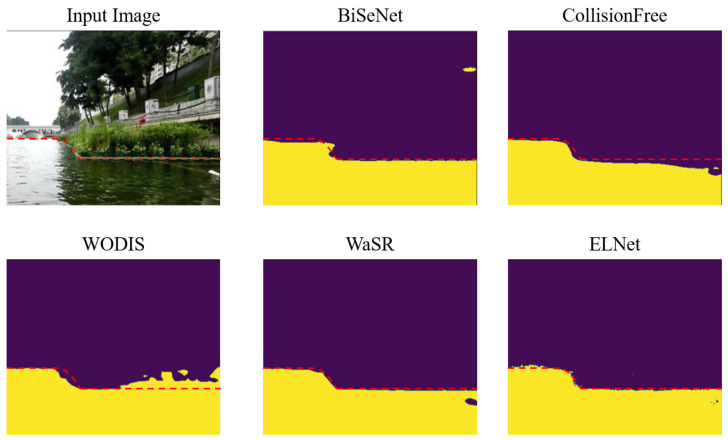
Qualitative evaluations on the USVINland dataset. The sky/obstacles and water are colored in purple and yellow, respectively. The red dashed line indicates the ground truth river edge.

**Table 1 sensors-23-04789-t001:** The architecture of the proposed network ELNet. “T” is the expansion factor of the bottleneck, “N” is the repeat times, “S” is stride, “OS” is the output scale of coding features, “IC” is input channel, “MC” is middle channel, and “OC” is output channel.

	Operator	OS	T	N	S	IC	MC	OC
Enc	conv2d	S/2	-	1	2	1	-	32
	bottleneck	S/2	1	1	1	32	-	16
	bottleneck	S/4	6	2	2	16	96/144	24
	bottleneck	S/8	6	3	2	24	144/192	32
	bottleneck	S/8	6	4	2	32	192/384	64
	bottleneck	S/16	6	3	1	64	384/576	96
	bottleneck	S/32	6	3	2	96	576/960	160
	bottleneck	S/32	6	1	1	160	960	320
Dec	up block	S/8	-	1	4	320	-	256
	up block	S/2	-	1	4	448	-	64
	conv2d	S	-	1	2	160	-	2
Aux	conv block	S/8	-	1	1	3	-	16
	conv block	S/8	-	1	1	192	-	4

**Table 2 sensors-23-04789-t002:** The number of trainable parameters and inference speed of compared segmentation methods.

Algorithms	Number of Parameters (M)	FPS on Gpu	FPS on Cpu
BiSeNet [39]	13.42	36.12	4.87
WODIS [9]	49.07	33.56	1.81
CollisionFree [10]	100.36	9.89	0.21
Skip-ENet [12]	0.75	59.82	11.24
WaSR [13]	71.50	10.63	0.98
ShorelineNet [14]	6.50	49.02	6.33
ELNet (Ours)	4.86	45.21	6.95

**Table 3 sensors-23-04789-t003:** Quantitative evaluation of segmentation methods on MODS benchmark in terms of water-edge localization accuracy ($\mu_{A}$) and detection robustness ($\mu_{R}$) with the percentage of correctly detected water edge pixels in parentheses, with precision (Pr), recall (Re) and F-score in percentages.

Algorithms	$\mu_{A}[\mathrm{px}]\;(\mu_{R})$	TP	FP	FN	Pr [%]	Re [%]	F-Score [%]
BiSeNet [39]	12 (98.4)	51,045	33,152	1443	60.6	97.3	74.7
WODIS [9]	18 (97.1)	49,966	87,651	2522	36.3	95.2	52.6
CollisionFree [10]	53 (91.7)	45,528	14,797	6960	75.5	86.7	80.7
Skip-ENet [12]	25 (95.8)	48,786	178,013	3702	21.5	92.9	34.9
WaSR [13]	11 (98.6)	51,607	85,374	881	37.7	98.3	54.5
ShorelineNet [14]	12 (98.4)	49,643	131,130	2845	27.5	94.6	42.6
ELNet (Ours)	11 (98.5)	51,429	29,318	1156	63.7	97.8	75.1

**Table 4 sensors-23-04789-t004:** Quantitative Evaluation of segmentation methods on USVInland dataset in terms of precision (Pr), recall (Re) and F-score in percentages.

Algorithms	Pr [%]	Re [%]	F-Score
BiSeNet [39]	97.89	87.77	93.04
WODIS [9]	96.14	88.84	92.68
CollisionFree [10]	93.64	75.96	84.88
Skip-ENet [12]	96.81	76.78	86.65
WaSR [13]	97.96	83.23	90.55
ShorelineNet [14]	96.60	81.17	88.83
ELNet (Ours)	97.82	88.69	93.96

**Table 5 sensors-23-04789-t005:** Ablation study of the upsampling number in terms of water-edge localization accuracy ($\mu_{A}$) and detection robustness ($\mu_{R}$), precision (Pr), recall (Re) and F-score in percentages.

	Params	$\mu_{A}[\mathrm{px}]\;(\mu_{R})$	Pr [%]	Re [%]	F-Score [%]
1	8.34 M	14 (98.0)	60.6	95.9	64.7
2	4.42 M	28 (95.3)	20.9	92.3	34.3
3	4.87 M	11 (98.5)	63.7	97.8	75.1
4	6.05 M	10 (98.6)	63.4	97.7	75.2
5	5.17 M	14 (98.0)	60.2	96.6	69.5

**Table 6 sensors-23-04789-t006:** Ablation study of edge-aware modules on USVInland.

Fusion	Auxiliary	F-Score	Re [%]	Pr [%]
×	×	96.96	88.12	93.99
✓	×	97.32	88.37	93.97
×	✓	97.36	88.42	93.88
✓	✓	97.82	88.69	93.96

## Data Availability

The original dataset MaSTr1325 and MODS benchmark can be obtained online by Bovcon et al. [41,42]. The USVInland dataset can be obtained from the access online by Cheng et al. [43].

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
