# Peer review of "Towards Smaller and Stronger: An Edge-Aware Lightweight Segmentation Approach for Unmanned Surface Vehicles in Water Scenarios"

_sensors, 2023, doi:10.3390/s23104789_

Round 1
Reviewer 1 Report (Previous Reviewer 3)
The suggested revisions have been made, and the paper is suitable for publication.
No further comments.
Author Response
Thanks for reading our paper carefully and giving the above positive comments. As for editing of language, we have associated a native English-speaking colleague to check our manuscript. The manuscript has been well checked.
Reviewer 2 Report (Previous Reviewer 2)
This revision is well prepared, and fixes some errors occurred in last manuscript. I think it could be accepted.
Author Response
Thanks for reading our paper carefully and giving the above positive comments.
Reviewer 3 Report (New Reviewer)
This paper presents an edge-aware lightweight segmentation approach for USV in water scenarios. In the proposed approach, the authors put forward a new architecture with two-stream learning and edge prior for water scenario segmentation. The weight of the new architecture is lighter, and the experiment conducted by the authors prove that the performance of the model has been improved as compared with other past architectures. The reviewer feels the method proposed by the authors has some novelty and this method can contribute to solve the problem existing in current study. However, the problem that the authors want to solve should be demonstrated explicitly in the paper and the references should be simply introduced to indicate why they triggered the authors’ idea. Furthermore, the English must be improved to be accepted by the journal.
Comments:
First the English must be improved. A review by a native speaker or service is needed. The motivation in the introduction is not expressed clearly.
1. In the Introduction section, the authors listed several references for previous research. However, the reviewer feels these references are not sufficient to describe the current status and difficulties in the application and the field. The motivation is thus not well identified at all. The situations in which this is needed are superficially discussed. Why are water applications different? Previous methods and attempts to deal with this problem are just not introduced in any detail. For this reason, the reviewer doesn’t clearly understand the current state of this field and why the authors’ study is necessary based on the description in the paper. Please explicitly demonstrate those points in the Introduction section. This is critical to getting the paper accepted.
2. The reviewer thinks the references in the Introduction section are not introduced in an explicit way. For example, the authors mention that their ideas are inspired by reference 13-15. What specifically is introduced in reference 13-15? Why do those inspire the authors’ idea? The reviewer expects that they will introduce the relevant content from those references. It is not well done at all.
3. In section 2.1 Network Architecture, the authors introduced the Laplacian Operator and the parameters set for this work. Please add an explanation of why the chosen parameters were selected and used here? What analysis was performed to make this determination? Are the values of those parameters generally suitable for all cases?
4. In section 2.1 Network Architecture, the authors mention that features from some stages are given up, which is meant to reduce the total number of parameters. Please add an explanation for why those features can be given up and how this action influences the results?
5. The authors compared the performance and inference speed of different segmentation methods. Is the performance shown in the paper only for MODS benchmark? Is it possible that for a different dataset, the results will be different?
6. The paper introduces enhancements to both performance and speed. The reviewer is requesting an explanation for how the proposed approach affects the training speed.
7. Tables and Figures should appear after they are first called in the paper.
First the English must be improved. A review by a native speaker or service is needed.
Author Response
Please see the attachment.

Reviewer 4 Report (New Reviewer)
Dear Authors,
Appreciate your efforts on discussing on one of important issues of recent automation application. Herewith i shared my thoughts on current version manuscript. Please find file attachment.

At some instances in the manuscript, language can be further modify with more better phrase. I would suggest for proof reading with native speaker.
Round 2
Reviewer 3 Report (New Reviewer)
Summary:
The reviewer believes that the modifications and responses have adequately addressed most of the concerns raised. The content is now more comprehensive and explicit. However, there is still room for improvement in the English language, and the paper should be modified accordingly based on the following minor comments. The reviewer recommends accepting the paper once the English has been improved and the following comments have been addressed.
Comment 1:
Regarding the response in Point 2, the reviewer believes that the trend of using autonomous vehicles should not be the primary motivation for conducting this study. However, focusing on the safety benefits of autonomous sailing is a valid point. Therefore, please adjust the content accordingly.
Comment 2:
Regarding the response to point 4, could you please provide a description or reference to support why the selected filter mask is applied to almost all cases for edge detection? Are there any previous studies that have demonstrated its effectiveness?
Comment 3:
The authors should provide a more comprehensive explanation of the variables in the equation. Some of the variables are not clearly defined in the paper, e.g., fea_i.
There is still room for improvement in the English language.
Author Response
Please see the attachment.

Reviewer 4 Report (New Reviewer)
Dear Authors,
Appreciate your efforts in improving manuscript in substantially and reflecting all the concerns highlighted in the previous revision.
And regarding response to concern 5 highlighted in the previous revision is still not convincing. As it highlighted in the conclusion proposed method achieves visible margin on the comprehensive accuracy of segmentation. Provided evidence are not conclusive to support this claim. Any figures highlights clear improvements of proposed method over the state of art method in terms of accuracy, would be recommended.
Author Response
Please see the attachment.

This manuscript is a resubmission of an earlier submission. The following is a list of the peer review reports and author responses from that submission.
Round 1
Reviewer 1 Report
A new architecture for water scenario segmentation is proposed in this paper. The authors introduce a light-weight network with two-stream learning and edge information in segmentation that can run on embedded devices for practical development of the field. The proposed scheme can achieve stable performance and real-time capacity.
Suggestions:
1. There are some spelling mistakes: e.g. in P2, "we are conscious of its importance of in raw image and ...” etc..
2. The experimental results need better explanations: e.g. in P11, Ablation Study on Edge-aware Modules, there should be more analysis to reveal the reason of the result shown in Table 5 & 6.
Reviewer 2 Report
This paper studies the water area segmentation problem in water scenarios. This problem is very useful for Unmanned Surface Vehicles. It considers the edge-fusion strategy and detail guidance and stablish a lightweight and well performed approach for the studied problem. These two improvements contributes to a lighter yet better network with lower computation. Experimental results show that the proposed method outperforms the latest existing work. The problem studied in this paper is significant, the paper is well written and easy to understand.
Some typos such as:
1. line 9, "...USVInland demonstrate the proposed model ELNet achieves..."->"...demonstrate that..."
2. lin 343, "This is to be improved point to alleviate in the future."
Reviewer 3 Report
The abstract and the conclusion section must be rewritten, highlighting the main results and the gap of knowledge.
In this research, a comparison with other methods and your modified ELNet approach has been evaluated. Can you also evaluate the ELNet approach performance?
Moreover, can you add the loss function results?
